# Near Perfect Absorber for Long-Wave Infrared Based on Localized Surface Plasmon Resonance

**DOI:** 10.3390/nano12234223

**Published:** 2022-11-27

**Authors:** Leihao Sun, Dingquan Liu, Junli Su, Xingyu Li, Sheng Zhou, Kaixuan Wang, Qiuyu Zhang

**Affiliations:** 1Shanghai Institute of Technical Physics, Chinese Academy of Sciences, Shanghai 200083, China; 2School of Physical Science and Technology, ShanghaiTech University, Shanghai 200031, China; 3School of Optoelectronics, University of Chinese Academy of Sciences, Beijing 100049, China

**Keywords:** near perfect broadband absorber, long-wave infrared, meta-structure, localized surface plasmon resonance

## Abstract

In recent years, broadband absorbers in the long-wave infrared (LWIR) spectrum have shown great scientific value and advantages in some areas, such as thermal imaging and radiation modulation. However, designing a broadband absorber with an ultra-high absorption rate has always been a challenge. In this paper, we design a near perfect absorber that is highly tunable, angle insensitive, and has polarization independence for LWIR. By using multi-mode localized surface plasmon resonance (LSPR) of a surface metal structure, the absorber achieves a very high absorption average of 99.7% in wavelengths from 9.7 μm to 12.0 μm. For incident light, the meta-structure absorber exhibits excellent polarization independence. When the incident angle increases from 0° up to 60°, the absorption rate maintains over 85%. By modulating the size of the structure, the meta-structure absorber can also achieve a high absorption rate of 95.6%, covering the entire LWIR band (8–14 μm in wavelength). This meta-structure absorber has application prospects in infrared detecting, infrared camouflage, radiation cooling, and other fields.

## 1. Introduction

Electromagnetic waves with wavelengths from 8 μm to 14 μm can usually be called long-wave infrared (LWIR) [1]. The electromagnetic waves in this band can efficiently avoid the absorption of atmospheric components such as water, carbon dioxide, and ozone. As such, LWIR is one of the main “atmospheric windows” [2]. In addition, LWIR is highly coincident with the wavelength of electromagnetic waves radiated by the human body, and thus has become the main band of infrared thermal imaging [3]. In recent years, based on some special properties of LWIR, sensing technology for this band has been widely studied and applied in medical thermal imaging [4], infrared stealth [5], bank note anti-counterfeiting [6], and meteorological monitoring [7]. In practical applications, it is needed to enhance the signal received by the infrared sensing device to avoid being drowned in the background noise. Such devices also have certain polarization and incident angle dependencies. A broadband absorber is the key to solving the problem. Absorbers with suitable bandwidth and high absorption efficiency can effectively improve the signal intensity and signal-to-noise ratio of infrared sensing signals [8]. In addition, the absorbers with a weak dependence on angle and polarization of incident light can help infrared sensors to receive optical signals in a wider angular range and with various polarization states [9]. These broadband absorbers also show excellent potential in infrared stealth [10], radiation cooling [11], and other fields.

The broadband absorbers have various forms such as simple multi-film, multi-layer array, nanoparticle filling, and meta-surface array. In 2014, Raman achieved absorption close to 70% in the band of 8–12 μm by using a multi-film structure [12]. The absorption of multi-film depends on the intrinsic absorption of the film materials and the interference effect between adjacent interfaces, so it is difficult to improve the absorption efficiency and absorption spectrum width perfectly. In the same year, R. Feng designed 2 × 2 multi-layer arrays of different sizes which achieved more than 90% absorption in the band of 8–12 μm in simulation calculation [13]. In 2020, Y. Luo used 3 × 3 multi-layer arrays to achieve more than 90% absorption in the band of 8–14 μm in a simulation calculation [14]. The requirement of the etching depth for such multi-layer arrays is too difficult to prepare under practical conditions. In 2016, L. Zhou achieved an average absorption of up to 99% in the band of 0.4–10 μm by filling the hole-like films with nanoparticles of metal [15]. In 2017, Y. Zhai filled SiO_2_ nanoparticles into the polymer to achieve more than 70% absorption in the band of 2.5–25 μm [16]. Although the nanoparticle filling method can achieve high absorption in a wide wavelength range, its film thickness is much higher than other types and is difficult to fabricate. Currently, the most widely researched method is meta-surface arrays. In 2019, Y. Luo modulated the surface metal structure based on the multi-layer array and achieved more than 80% absorption in the band of 8–14 μm [17]. In 2021, the surface metal cross structure designed by Y. Zhou achieved an average absorption of 95% in the band of 8–14 μm [18]. In 2022, the surface metal ring 2 × 2 array designed by Z. Qin achieved an average absorption of 93.8% in the band of 7.76–14.0 μm [19]. T. Xie’s and Y. Liang’s designs adopted two more complex surface metal structures. The former achieved more than 90% absorption peaks in the two atmospheric windows, respectively [20], and the latter achieved 94.3% of the average absorption in the band of 12.3–32.0 μm [21]. Meta-surface structure arrays can precisely control the absorption spectrum range by adjusting the size and morphology of surface metal structures, which are suitable for different application scenarios. With the development of surface nano-fabrication technology, some complex surface metal structures can also be fabricated by electron beam etching, nano-imprinting, and other methods [22]. Although the above-mentioned meta-surface structure array can realize the broadening of the absorption spectrum, the wavelength range in which the absorption rate can reach 99% is limited, which is far from meeting the requirements for perfect absorption. It is still a difficult challenge to design a broadband absorber with an absorption rate of 99% or more in a wide spectral range. 

Most of the above-mentioned meta-surface structure arrays are prepared from metals and metal-like materials, since metal micro-nano structures can support an important free-electron collective oscillatory behavior, namely surface plasmon polariton (SPP) [23]. This behavior occurs at the interface between the metal and the dielectric, which causes the electric field of the incident light to propagate along with the interface or be localized in the metal micro-nano structure. SPP can be divided into two categories, depending on the mode of propagation: those propagating along the interface are propagating surface plasmon (PSP) [24], while those being localized are localized surface plasmon (LSP) [25]. In general, the energy of the incident light cannot be coupled directly and effectively into the PSP because the incident light wave vector cannot meet the phase-matching condition with the PSP wave vector. This requires us to use some special methods to increase the component of the incident light wave vector in the direction of PSP propagation, which to some extent brings difficulties to the design and fabrication [24,25,26,27,28]. However, for LSP, the scattering effect of metallic micro-nano structured surfaces on the incident light field gives them a wide wave vector value distribution, so they can be directly excited by the incident light. LSP is a type of non-propagating surface wave formed by the mutual coupling of incident light and metal micro-nano structures. Since the metal structure surface can provide an effective restoring force for the collective oscillating electrons driven by the optical field, a field enhancement maximum can be generated near the surface of the metal nano-structure at a specific frequency, which is called localized surface plasmon resonance (LSPR) [24]. In 1908, Mie proposed the Mie theory, which expressed the optical response around metal micro-nano spheres by dipole approximation [29]. In 1912, Gans proposed the Gans–Mie theory based on the Mie theory, which approximated the response electric field at the periphery of the ellipsoid as three pairs of dipoles along different axes, and explained why there are multiple LSPR modes in the metal ellipsoid [30]. The resonant frequency of the LSPR is closely related to the size of the metallic structure. In the process of increasing the size of the metal structure, the distance of the heterodyne charge interaction increases, and the oscillating electron reply to coefficient of the response decreases, which makes the LSPR resonance frequency decrease (resonance wavelength red shift) [31]. The increase of radiation damping also makes the resonance spectral line broaden [32]. For a complex surface metal structure, multiple LSPR modes with different frequencies, different spectral line widths, and different resonance intensities may exist simultaneously [33]. High absorption over a wide spectral range can then be achieved by the integration of multiple LSPR modes.

In this paper, a meta-structure absorber based on LSPR is proposed, which can maintain near-perfect absorption of more than 99% over a broad spectral range of 9.7–12.0 μm, with an average absorption rate of 99.7%. The absorption spectrum and effective impedance of the structure are calculated to indicate the macroscopic optical properties. In order to further research the physical mechanism of high absorption at microscopic level, we simulated the electric field on the surface of the structure. The results show that high absorption is achieved by the combined effect of six types of LSPR modes, and the secondary response between different LSPR modes. For these modes, we investigated the effects of seven structural parameters on the absorption spectra, respectively. We also change the polarization state and the incident angle to find that the absorber exhibits excellent polarization independence and insensitivity to the incident angle. Finally, the absorber achieved an average absorption of 95.7% in the band of 8–14 μm by tuning the surface structure and the dielectric layer structure, which proves that our absorber is tunable. Our meta-structure absorber is simple, tunable, slightly polarization independent, and angle insensitive, which can be applied in infrared thermal imaging, infrared stealth, radiation cooling, and other fields.

## 2. Structure and the Simulation Methods

For a near perfect absorber in the band of 8–14 μm spectral, we designed a meta-structure absorber. The structure is shown in Figure 1. The absorber can be decomposed into three layers of metal–dielectric–metal. The uppermost layer is a meta-surface of micro-structure array prepared by Ti, and its thickness is labeled as T; the middle dielectric layer is composed of Ge and TiO_2_, and its thickness is labeled as D and H, respectively; the bottom metal layer is a whole layer of Ti, and the thickness is labeled as L.

Figure 1a shows a schematic three-dimensional structure of the meta-structure absorber. Figure 1b shows the top view of a representative period in the metal surface structure; the period is labeled as P. The structure is a centrosymmetric structure composed of a central ring and four semi-elliptical rings of the same size. For the ring, its inner and outer diameters are characterized by R_1_ and R_2_. For the elliptical ring, its inner and outer long axes are characterized by A_1_ and A_2_; its inner and outer short axes are characterized by B_1_ and B_2_. The top layer of Ti with a thickness of 20 nm is used to excite the LSPR effect. The second layer of Ge with a thickness of 250 nm is a unique semi-metal. It is far more conductive than non-metals and can provide an environment for the migration of charges in the LSPR effect. At the same time, its refractive index is much lower than other metals, which can effectively reduce the reflection from the surface. The third layer of TiO_2_ is used as an absorber layer. We found a very significant increase in the k of TiO_2_ in the 8–14 μm interval [34], which implies a significant absorption in the long-wave infrared region of the material. The addition of a layer of TiO_2_ with a thickness of 490 nm has a significant effect on the enhancement of absorption. The bottom layer of Ti is 120 nm thick, which is larger than its skinning depth and acts similar to a highly reflective mirror. The Ti (20 nm)-Ge (250 nm)-TiO_2_ (490 nm)-Ti (120 nm) four layers can be prepared by electron beam evaporation or magnetron sputtering. For a small area, this meta-surface can be prepared by focused ion beam etching. If the structure needs to be prepared over a large area, the nanoimprint lithography method can be considered.

In this paper, the finite difference time domain method (FDTD) is used to simulate and calculate the optoelectronic characteristics of the absorber. The physical parameters of Ge and Ti used in the calculation are from E. D. Palik [35], and the physical parameters of TiO_2_ are from J. Kischkat [34]. Light reaches the surface along the negative direction of the z-axis, and the polarization direction is along the x-axis. Periodic boundary conditions are used in the x and y directions, and the perfectly matched layer (PML) boundary is used in the z-direction. The specific simulation parameters will be given in the first part of the Appendix A. After optimization, the geometrical parameters of the absorber are set as T—20 nm, D—250 nm, H—490 nm, L—120 nm, P—2000 nm, R_1_—220 nm, R_2_—400 nm, A_1_—500 nm, A_2_—800 nm, B_1_—200 nm, and B_2_—460 nm. The total thickness of the meta-structure absorber is 880 nm, less than one-tenth of the central wavelength of the absorption peak.

## 3. Results and Discussion

The absorption (A) spectrum of the absorber is calculated from the reflectance (R) and transmittance (T) obtained by simulation, A = 1−R−T. The solid red line in Figure 2b shows that the superstructure absorber can maintain absorption more than 99% in the wavelength range of 9.7–12.0 μm, with an average absorption rate of 99.7%. The solid black line in Figure 2a is the absorption spectrum of the absorber without the surface structure. It is found that the high absorption mechanism of the absorber is mainly dependent on the surface metal structure. To explain the physical mechanism of broadband absorption at the macro level, we retrieved the effective impedance of the meta-structure absorber by the S-parameters method [36], which is expressed by:(1)$$Z=\sqrt{\frac{\mu }{\varepsilon }}=\sqrt{\frac{{\left(1+S_{11}\right)}^{2}-S_{21}^{2}}{{\left(1-S_{11}\right)}^{2}-S_{21}^{2}}}$$
where *μ* and *ε* represent the effective permeability and effective permittivity of the device, respectively. *S*_11_ and *S*_21_ are physical quantities that characterize the reflection and transmission characteristics of the absorber. The specific use of S-parameters will be given in the second part of the Appendix A. Figure 2c shows the real part and the imaginary part of the impedance calculated with S-parameters. The result shows that the real part (solid blue line) is close to 1 and the imaginary part (solid red line) is close to 0 in the range from 9.7 μm to 12.0 μm. It means that the impedance of the absorber matches that of the free space well and meets the requirements of designing a perfect absorber.

To explain the physical mechanism of perfect absorption at the microscopic level, we calculated the electric field distribution in the interface between the top Ti array and the Ge layer. Figure 3 shows the electric field distribution corresponding to 8–13 μm incident light (TM wave). When the incident light wavelength is 8–9 μm, the surface electric field is mainly distributed in the inner ring, the outer ring of the central ring, and the inner ring of the elliptical ring in the y-axis direction. The electric field in the inner and outer ring regions of the central ring increases obviously with the increase of wavelength. When the wavelength of the incident light is 9–12 μm, the electric field is distributed in the inner and outer rings of the central ring and the inner and outer rings of the elliptical ring. Among them, the electric field in the inner and outer ring regions of the central ring is obviously weakened with the increase of wavelength, while the electric field in the inner and outer ring regions of the elliptical ring is obviously increased with the increase of wavelength. When the incident light wavelength is 12–13 μm, the electric field is distributed in the inner and outer rings of the central ring and the inner and outer rings of the elliptical ring. The electric field strength is weakened to different degrees in each region.

Then, we analyzed the LSPR modes in the structure. This meta-surface mainly has metal structures in two shapes: a circular ring and elliptical ring. As shown in Figure 3, when the wavelength of the incident light is 8–9 μm, there is an electric field in the inner ring area of the central ring with moderate intensity, which indicates that the first LSPR mode is dominated by the inner ring of the ring. When λ = 8.47 µm, the electric field intensity in this region reaches its maximum, denoted as λ_1_. When the incident light wavelength is 8–13 μm, there is also a significant electric field distribution in the outer ring region of the central ring, which indicates that the second LSPR mode is dominated by the outer ring of the ring. When λ = 8.99 µm, the electric field intensity in this region reaches its maximum, denoted as λ_2_. When the wavelength of the incident light is 10–12 μm, there is a high-intensity electric field distribution in the inner ring region of the elliptical ring, which is the third LSPR mode, denoted as λ_3_. When the wavelength of incident light is between 11–13 μm, there is a high-intensity electric field distribution in the outer ring region of the elliptical ring, which is the fourth LSPR mode, denoted as λ_4_. Due to the asymmetry of the elliptical ring in the x and y directions, the LSPR mode cannot be simply divided into two modes. It can be seen that as the incident light wavelength increases, the response electric field in the central annular region moves from the inner annular region to the outer annular region. The response electric field of the surrounding elliptical ring area moves from the inner ring area to the outer ring area, while also shows a trend of moving from the short axis direction to the long axis direction. This performance shows that the two LSPR modes λ_3_ and λ_4_ corresponding to the inner and outer rings of the elliptical ring can be further decomposed into four modes, denoted as λ_31_, λ_32_, λ_41_, and λ_42_; corresponding to the direction of the long axis of the inner ring of the elliptical ring, the direction of the short axis of the inner ring of the elliptical ring, the direction of the long axis of the outer ring of the elliptical ring, and the direction of the short axis of the outer ring of the elliptical ring, respectively. Similarly, by finding the wavelength at which the electric field strength is maximum, we obtain λ_31_ = 9.92 µm, λ_32_ = 10.46 µm, λ_41_ = 11.97 µm, and λ_42_ = 12.69 µm. To investigate the specific effect of these modes on the absorption, we calculate the absorbance curves for different shapes in Figure 4.

Figure 4a shows the absorption spectra of different circular structures. There is a slight increase in the absorption rate as the shape changes from a small circle to a large circle, which is due to the size sensitivity of LSPR. For the circular ring, in the above electric field analysis, we found that the inner and outer ring regions of the circle have strong electric field distributions at wavelength λ_1_ and wavelength λ_2_, respectively. We speculate that the electric field distribution in the outer ring region is similar to that of a large circle, while the electric field distribution in the inner ring region is similar to that of a small circle. In this way, we can approximate the absorption of the circular ring as a hybridisation of the absorption of the small and large circles. This speculation can also be applied to elliptical structures. It should be noted that the two insignificant absorption peaks in the absorption pattern do not correspond exactly to λ_1_ and λ_2_ due to the absorption of the two dielectric layers. Combined with the electric field analysis, the circular structure makes an increase in the absorption in the wavelength range of 8–10 µm. In Figure 4b, there is an obvious increase in the absorption rate as the shape changes from a small ellipse to a large ellipse. The absorption spectrum of the large ellipse shows two very obvious absorption peaks, which, combined with the analysis of the electric field distribution mentioned above, can correspond to the short and long axes of the ellipse, respectively. At the same time, the resonance wavelength of LSPR is red-shifted as the ellipse size increases, consistent with the mainstream LSPR theory. In the same way, we can approximate the absorption of the ellipse ring as a hybridisation of the absorption of the small and large ellipse. It is important to note that, in addition, to the short and long axis regions of the elliptical ring, there are also a large number of dispersed charge distributions in other nearby regions. In such a situation, it is very difficult to find the absorption peaks corresponding to λ_31_, λ_32_, λ_41_, and λ_42_. Figure 4c,d shows the impact of the composite structure, which broadens the absorption spectrum and enhances the absorption to near perfection. On the one hand, the central ring enhances the absorption in the 8–10 µm range by its own absorption. On the other hand, a secondary response occurs between the circular ring and the LSPR excitation electric field of the adjacent elliptical ring. As can be seen in Figure 3, in addition to the inner and outer ring regions of the circular and elliptical rings, electric fields of lower intensity are also present in the region between the adjacent elliptical rings and the region between the adjacent circular and elliptical rings. Overall, the resonance frequencies, resonance intensities and resonance peak linewidths corresponding to these six modes are partially different, but they are complemented by integration and mutual coupling in the spectral range of 8–13 μm, which finally enables ultra-high absorption of up to 99.7% to be achieved in the broad spectral range of 9.7–12.0 μm.

Considering the correlation between the meta-surface structure size and the properties of LSPR, we explored the influence of the structure size parameters on the absorption. The six LSPR modes in the absorber can be corresponded to the six dimensional parameters: R_1_, R_2_, A_1_, A_2_, B_1_, and B_2_, respectively. In addition, the structural period (P) is the seventh factor to be considered due to the coupling effect of adjacent electric fields. Figure 5, Figure 6 and Figure 7 show the effects of the seven parameters on the absorption spectra. Only a unique variable exists for each parameter study, and the solid blue lines in all figures represent our standard structures.

Figure 5a shows the absorption spectra of different structural periods. When P is less than the standard, the absorption spectral range become wider as P decreases, and the absorbance in the working wavelength decreases significantly. When P is larger than the standard, the absorption spectral range become narrower as P increases, and the absorbance in the central working wavelength reaches the extreme value at *p* = 2.2 μm, and then shows a decreasing trend. For a better illustration, we need to use three parameters (d_1_, d_2_, d_3_) in Figure 5b. When *p* = 1.6 μm, we find two distinct absorption peaks and define them as short-wave peak and long-wave peak. As the distance d_1_ between the ring and elliptical ring decreases, the coupling effect between them becomes stronger. Similarly, as the distance d_2_ and d_3_ between different elliptical rings decreases, the coupling effect between them becomes stronger, too. These effects cause more charges to accumulate in the short and long axis directions of the elliptical ring, as well as in the outer ring region of the ring. The small size of the outer ring of the circle and the short axis of the elliptical ring makes the short-wave peak appear in the absorption spectrum, and the larger size of the long axis of the elliptical ring makes the long-wave peak appear. As p increases, d_1_, d_2_, and d_3_ also increase, making the coupling effect weaker. When *p* = 2 μm, the more dispersed charge distribution achieves a delicate balance that allows the absorption over a wide wavelength range to be maintained at an extremely high level. As p continues to increase, the weaker coupling effect and the more dispersed charge distribution make the absorption region narrower, and the absorption rate decreases. According to this analysis, similar trends will appear in the spectra of different elliptical and circular ring sizes, which make changes to d_1_, d_2_, and d_3_.

The effects of the radii of the inner and outer rings of the central circle on the absorption spectrum are shown in Figure 6. In Figure 6a, increasing R_1_ causes a slight broadening of the absorption spectrum and a slight decrease in the absorption rate. In Figure 6b, increasing R_2_ causes a slight decrease in the absorption rate for the short-wave region and shifts the absorption spectrum toward the long-wave direction (red-shift).

Figure 7 shows the effects of the four parameters characterizing the elliptical rings on the absorption spectrum. In Figure 7a,b, the trends of absorption spectra with A_1_ and with B_1_ show a high degree of similarity. When A_1_ and B_1_ are smaller than their standards, with the decrease in size, the absorption spectrum is blue-shifted. Meanwhile the absorption rate decreases significantly while the decreasing extent of the short-wave region is greater than that of the long-wave region. When A_1_ and B_1_ are larger than the standard, with the increase in size, the absorption spectrum is red-shifted. Meanwhile, the absorption rate decreases significantly while the decreasing extent of the long-wave region is greater than that of the short-wave region. In Figure 7c, when A_2_ is smaller than the standard, the absorption spectral range becomes significantly narrower as A_2_ decreases, and the absorbance of the central working wavelength reaches the extreme value at A_2_ = 370 nm and then shows a decreasing trend. When A_2_ is larger than the standard, the absorption spectral range widens as A_2_ increases, and the absorbance within the working wavelength decreases significantly. In Figure 7d, when B_2_ is smaller than the standard, the absorption spectral range becomes narrower as B_2_ decreases, and the absorption rate decreases significantly. Conversely, the decreasing extent of the long wave region is greater than that of the short wave region. When B_2_ is larger than the standard, the absorption spectral range widens as B_2_ increases and the absorption rate decreases significantly while the decreasing extent of the short-wave region is greater than that of the long-wave region.

Depending on the mechanism, we can divide the trends of absorption into three types. The first type is entirely determined by the sensitivity of the different LSPR modes to the corresponding structure sizes. The parameters A_1_, B_1_, and R_2_ are consistent with this trend. The second type affects the absorber performance by the coupling effect of adjacent resonant electric fields. The parameter that fits this type is P. The third type includes both the sensitivity of LSPR to size and the effect of coupling effects between electric fields. The parameters A_2_, B_2_, R_1_ are consistent with this trend. The absorber we have designed reaches a balance between achieving a wider spectral range and having an ultra-high absorption (maintain more than 99%) by adjusting the combined effects of the seven parameters.

Considering the practical applications, the states of the incident light have also been discussed. Figure 8a shows the absorption spectra under different polarization states of the incident light. Due to the high symmetry of the meta-surface structure, the absorption spectra hardly change when the polarization angle gradually increases from 0° to 90°, showing polarization independence. Figure 8b,c shows the absorption spectra under different incident angles for transverse electric (TE) polarization and transverse magnetic (TM) polarization, respectively. In Figure 8b, when the incident angle of the TE wave reaches 60°, the absorber still achieves an average absorption of 93.2% in the working band (9.7–12.0 μm in wavelength). In Figure 8c, when the incident angle of TM wave reaches 60°, the absorber also achieves an average absorption of 87.8% in the working band. Therefore, the absorber shows angle insensitivity to some extent.

Finally, we discuss the possibility of adjusting the absorber’s working mode. For the absorber, there is a significant negative correlation between the bandwidth and the absorption rate. To meet the demand for absorbance and bandwidth in different applications, we divided the absorber into narrowband mode (bandwidth: 9.7–12.0 μm, absorption rate: maintained at 99% or more, average—99.7%) and broadband mode (bandwidth: 8–14 μm, absorption rate: maintained at 90% or more, average—95.6%). The adjustment from narrowband mode to broadband mode consists of two steps. In the first step, we adjusted the parameters of the surface structure, and the effect is shown in Figure 9a. Adjusted surface structure size: P-2000 nm, R_1_-200 nm, R_2_-460 nm, A_1_-460 nm, A_2_-1040 nm, B_1_-240 nm, and B_2_-520 nm. In the second step, we kept the total thickness constant and only replaced the TiO_2_ film layers with different layers of SiO_2_-TiO_2_ films (ST), and the effect is shown in Figure 9b. The thicknesses of the three film systems are as follows: (1) TiO_2_ (490 nm); (2) SiO_2_ (240 nm)-TiO_2_ (250 nm); (3) SiO_2_ (90 nm)-TiO_2_ (50 nm)-SiO_2_ (100 nm)-TiO_2_ (100 nm)-SiO_2_ (100 nm)-TiO_2_ (50 nm). Theoretical studies have shown that the reflected waves at the interface between the bottom and the top multi-layer undergo phase extinction, which result in the presence of absorption in the film, causing the absorption spectral range to be broadened [37]. As the number of layers of ST increases from 1 to 3, we can broaden the absorption spectrum to cover the entire long-wave infrared band, maintaining an absorption above 90% in the 8–14 μm wavelength range with an average absorption of 95.6%. Overall, the use of both methods allows the absorption bandwidth to be widened while maintaining a high absorption rate, achieving the broadband mode.

## 4. Conclusions

In summary, we designed a meta-structure absorber based on LSPR (localized surface plasmon resonance) theory for the long-wave infrared band (9.7–12 μm wavelength), achieving 99.7% near-perfect absorption. At the macroscopic level, we used impedance matching theory to verify the high absorption performance of the absorber. At the microscopic level, we analyzed the electric field on the meta-surface of the structure, and the physical mechanism of the ultra-high absorption in the broad band was explained. The results of the electric field distribution show that the ultra-high absorption is achieved mainly by the combined action of six LSPR modes. We analyzed seven parameters affecting the absorber performance, including six parameters corresponding to the six LSPR modes and the structural period parameter, which affects the coupling effect of different LSPR modes. For incident light, the meta-structure absorber exhibits excellent polarization independence. The absorber maintains an absorption rate of over 85% when the incident angle increases from 0° up to 60°. Finally, by adjusting the structural parameters and the multi-layer film, the absorber achieved an average absorption of 95.6% in the spectral range of 8–14 μm. The designed ultra-structured absorber has application prospects in the fields of thermal detection, infrared stealth, radiation cooling, and others.

## Figures and Tables

**Figure 1 nanomaterials-12-04223-f001:**
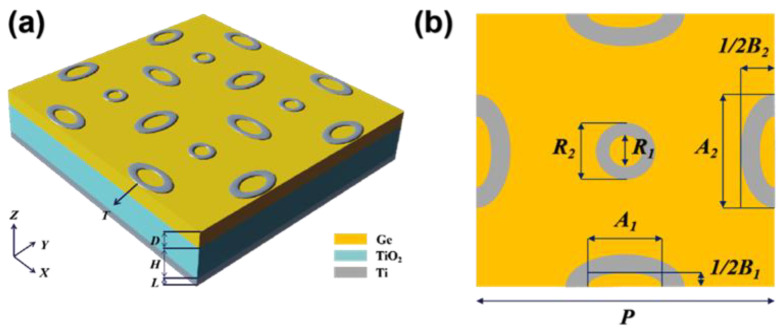
Schematic diagram of the designed meta-structure absorber. (**a**) Three-dimensional diagram includes a top array of Ti meta-structure and three layers of Ge, TiO_2_, and Ti. The geometric parameters used to characterize the thickness in the figure are T—20 nm, D—250 nm, H—490 nm, and L—120 nm. (**b**) One period of the Ti array. The geometric parameters used to characterize the size in the figure are P—2000 nm, R_1_—220 nm, R_2_—400 nm, A_1_—500 nm, A_2_—800 nm, B_1_—200 nm, and B_2_—460 nm.

**Figure 2 nanomaterials-12-04223-f002:**
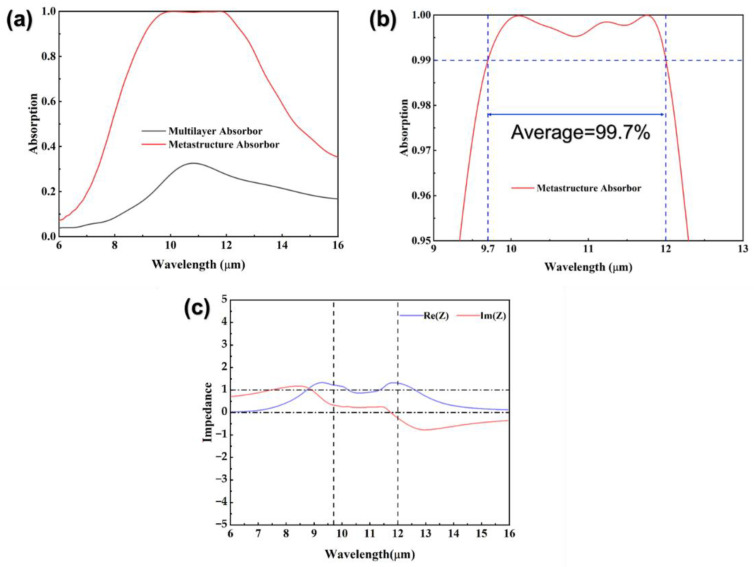
(**a**) Absorption spectrum diagram of the meta-structure absorber. (**b**) An enlarged view of the absorption spectrum. (**c**) Effective impedance of the meta-structure absorber in the working wavelength range.

**Figure 3 nanomaterials-12-04223-f003:**
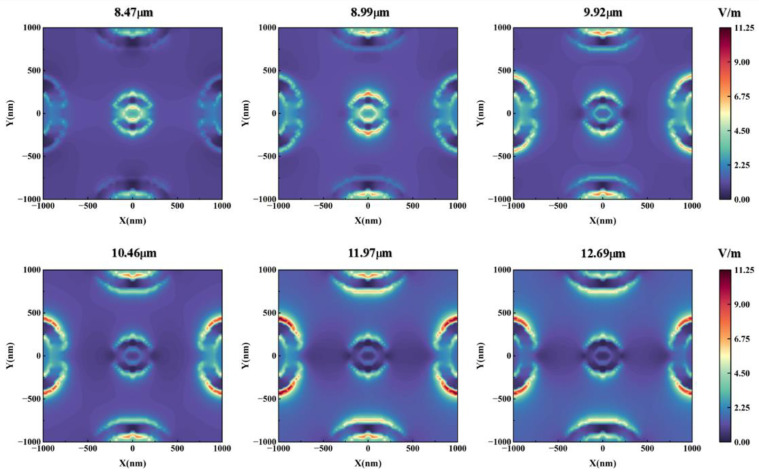
Surface electric field distribution under different wavelengths of incident light (TM wave).

**Figure 4 nanomaterials-12-04223-f004:**
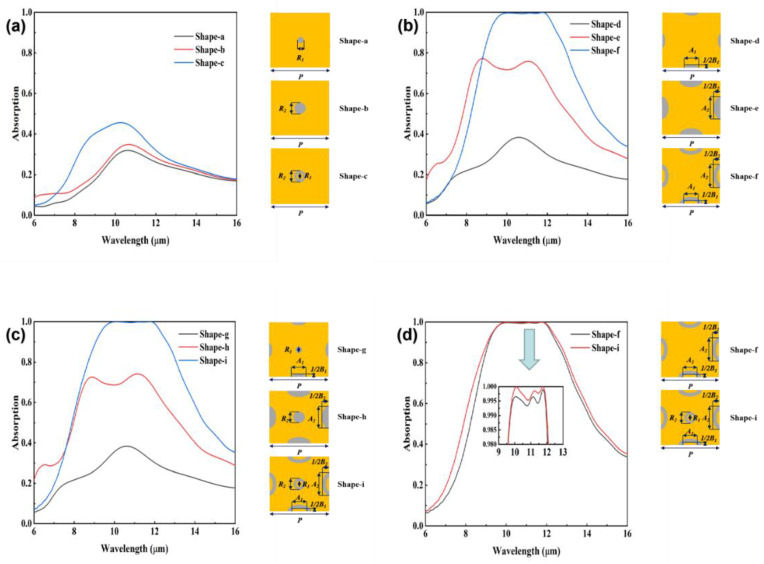
Absorption spectra of different shapes. These shapes are all 20 nm thick and have a period of 2 μm. (**a**) Circular structure. Shape-a: small circle, R_1_ = 220 nm. Shape-b: large circle, R_2_ = 400 nm. Shape-a: ring, R_1_ = 220 nm, R_2_ = 400 nm. (**b**) Elliptical structure. Shape-d: small ellipse, A_1_—500 nm, B_1_—200 nm. Shape-e: large ellipse, A_2_—800 nm, B_2_—460 nm. Shape-f: Elliptical ring, A_1_—500 nm, B_1_—200 nm, A_2_—800 nm, B_2_—460 nm. (**c**) Composite structure. Shape-g: small size, A_1_—500 nm, B_1_—200 nm, R_1_—220 nm. Shape-h: large size, A_2_—800 nm, B_2_—460 nm, R_2_—400 nm. Shape-i: Etched ring, R_1_—220 nm, R_2_—400 nm, A_1_—500 nm, A_2_—800 nm, B_1_—200 nm, and B_2_—460 nm. (**d**) Comparison with and without the central ring.

**Figure 5 nanomaterials-12-04223-f005:**
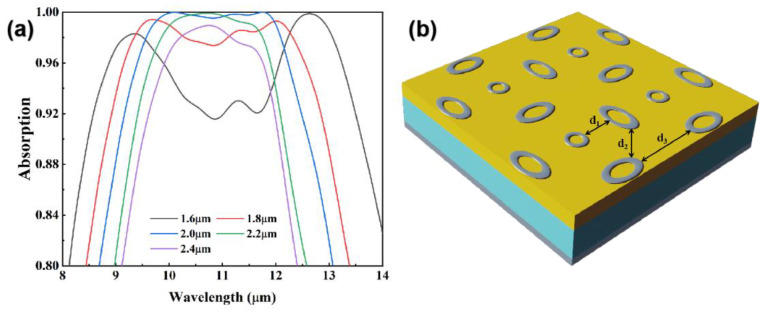
(**a**)Absorption spectra with different period sizes. (**b**) Schematic diagram of the designed meta-structure absorber. Three parameters: d_1_—distance between ring and elliptical ring; d_2_—distance between adjacent elliptical rings in different directions; d_3_—distance between adjacent elliptical rings in the same direction.

**Figure 6 nanomaterials-12-04223-f006:**
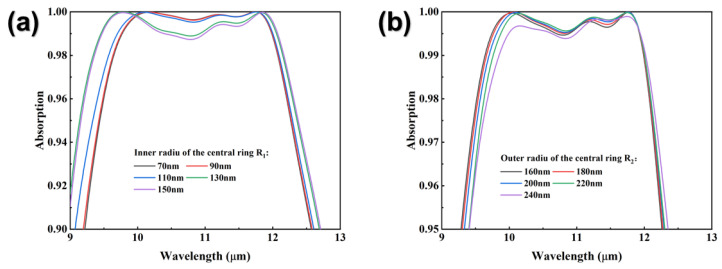
(**a**) Absorption spectra with different radii of the inner ring of the central circle. (**b**) Absorption spectra with different radii of the outer ring of the central circle.

**Figure 7 nanomaterials-12-04223-f007:**
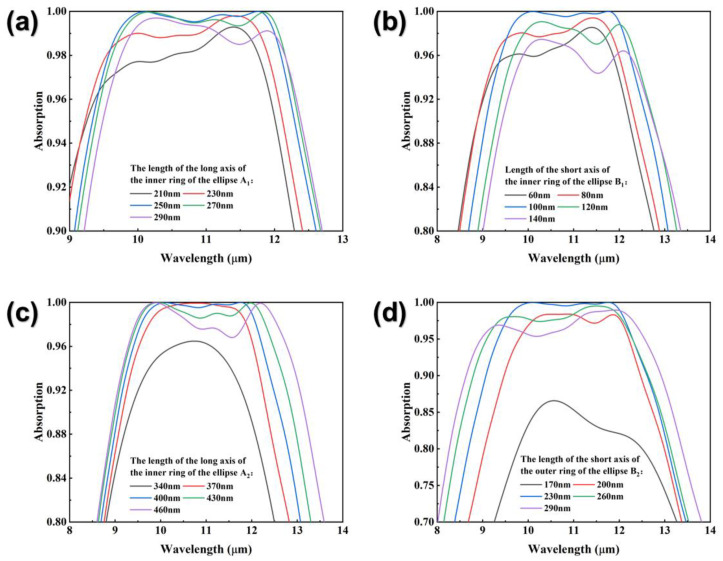
(**a**) Absorption spectra with different long axes of the inner ring of the elliptical ring. (**b**) Absorption spectra with different short axes of the inner ring of the elliptical ring. (**c**) Absorption spectra with different long axes of the outer ring of the elliptical ring. (**d**) Absorption spectra with different short axes of the outer ring of the elliptical ring.

**Figure 8 nanomaterials-12-04223-f008:**
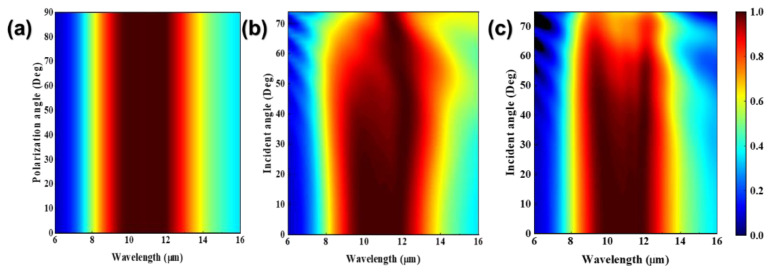
(**a**) Absorption spectrum under different polarization angles. (**b**) Absorption spectrum under different incident angles (TE wave). (**c**) Absorption spectrum under different incident angles (TM wave).

**Figure 9 nanomaterials-12-04223-f009:**
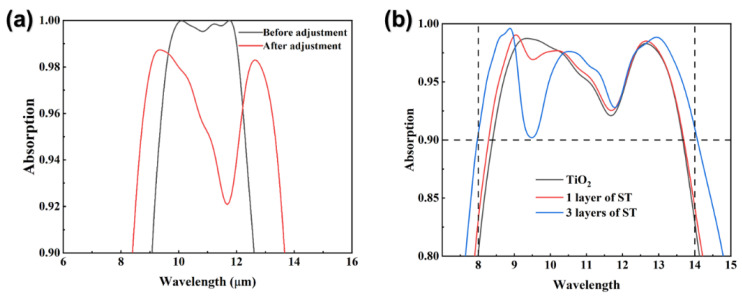
(**a**) Absorption spectra obtained by adjusting the surface structure. (**b**) Absorption spectra obtained by replacing the dielectric layer with ST (SiO_2_-TiO_2_ film) layers.

## Data Availability

Data underlying the results presented in this paper are not publicly available at this time, but may be obtained from the authors upon reasonable request.

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
