# Peer review of "Near Perfect Absorber for Long-Wave Infrared Based on Localized Surface Plasmon Resonance"

_nanomaterials, 2022, doi:10.3390/nano12234223_

Round 1

Reviewer 1 Report

The manuscript is presenting some simulation results aimed at show the performances of a plasmonic structure in obtaining an almost perfect absorber over a relatively wide spectral interval in the medium infrared.

The topic is interesting and, as reported in the introduction, suitable for possible applications. Nevertheless, the manuscript presents just the simulation model whitout any consideration or comment on the real possibility of realizing such kind of structure.

Unfortunately, the method used and/or the presentation are not properly supporting the claim of the manuscript.

No details are done on the FDTD simulation procedure (parametrization, convergence criteria, mesh, boundary conditions, ...); so, even a qualitative evaluation of the results is impossible.

Apparently, a matrix method for the calculation of the impedance has been applied, but no indication of the meaning and the origin of the used parameters is reported (where are the S11 and S21 parameters obtained from ?). By the way, the claim that the real part of the impedance (ranging roughly from 0.8 and 1.3) is close to 1 and the imaginary part (from -0.3 to 0.3) is close to 0, has to be better understood or justified.

A reiterated list of the main features occurring in the field map has been tentatively associated to LSPR modes, but no values have been assigned to the cited wavelengths.

A description of the absorbance dependence on the (7 !) parameters of the metastructure absorber and of the observed trends is presented, but just a generic interpretation is given. It lacks an analisys of the role played by the concurring LSPR modes and of their interaction. The repeated claim that an observed trend is "obvious" is not satisfactory.

How the parameter optimization procedure has been implemented and the optimized values obtained ?

It is not clear what is the role of the Ge, TiO2 and Ti layers below the metastructure and of their thickness. However, in the final part of the manuscript, the Absorbance spectra related to the substitution of the TiO2 layer with a not well defined stack of SiO2/TiO2 layers is shown, affecting the spectral range.

A "comparison between a narrow-band mode before adjustment" and a "broad-band mode after adjustment" is also reported, but it is not at all clear what kind of "adjustment" is intended (what is the starting point, what are the adjusted parameters and what has been the fitting function to be minimized).

As a conclusion, the manuscript appears not suitable for publication on "Nanomaterials", giving no physical insight into the mechanism leading to an efficient and realistic absorber device

Author Response

Dear Reviewer,

Thank you for your carefulness and conscientiousness. Your suggestions are really valuable and helpful for revising and improving our paper. For the details of the revisions we have explain them in a cover letter , please check it. 

We look forward to hearing from you at your earliest convenience.

Sincerely yours,

Dingquan Liu  PhD, Prof.

Reviewer 2 Report

Disclaimer: All comments are meant to be helpful. If I fail in doing so, I apologize in advance.

======================

The manuscript "Near perfect absorber for long-wave infrared based on localized surface plasmon resonance" by Leihao Sun , Dingquan Liu , Junli Su , Xingyu Li , Sheng Zhou , Kaixuan Wang and Qiuyu Zhang introduces an infrared meta-material with three plasmonic ellipsoid void resonators tuned to cover a broadband absorption >99% in the wavelength range from 9.7μm to 12.0μm.

The presentation of the research is straight forward and the results are convincing, however, the study is only performed numerically. Overall the research can be published after a few issues have been adressed:

I would like to see in Figure 2 the claim of being above 99,7% in absorption. Therefore, I would like to see an inset, a log-scaling or at least a reference to figure 5, where the claim is easier to see in the presented data (still not sufficiently detailed).

------------

I am missing simulations of the fundamental building blocks (ring and ellipse with and without enclosing goldlayer) and references to comparable literature as this would help to distinguish between the impact of the localized modes, their hybridisation and the couplings in the gold film (strongly visible by altering the period P). Also not mentioned is the hybridization which occur in these ring structures. So the physical interpretation of the results is only scratching the surface of the underlying fundamental effects.

-----------

I do not understand why the authors perform a 45° analysis as linear polarizations can be superposed linearily from both x- and y-polarized plane waves. This is seen in Fig. 8a. So the paper can be shortened significantly. This should be done to enhance the clarity of presentation.

All the best on improving this work.

Author Response

(The authors gave the same response as above.)

Round 2

Reviewer 1 Report

The authors took into serious consideration the observations made in the first review report. 

The largest part of the revealed flaws has been corrected and emended and the presentation is now more clear and effective.

In my opinion there is still a couple of questions which should be better defined.

I still find the discussion at page 5 in terms of effective impedance not fully convincing. It doesn't seem to add much in terms of understanding. Considering the relatively large spread of values of Re(Z) and Im(Z) around 1 and 0, respectively, shown in Figure 2(c), it could be more effective when the authors can extend the plot  between 6 and 16 um , like in Figure 2(a), or, at least, between 9 and 13 um , like in Figure 2(b).

About the discussion related to the "narrow-" and "broad-" band modes at pag.11, there is one point which should deserve a clearer statement. As far as can be understood, the reference "unadjusted" structure is the optimized one with the TiO2 layer, whereas the "adjusted" broad-band structure has the same surface plasmon parameters but the TiO2 layer substituted by the more complex multilayer structure. Is this correct ? Is the adjustment just related to the multilayer structure keeping the plasmonic features unchanged ? It should be better that this is explicitly stated.

I would be pleased when the authors would take into consideration  these two last observations. 

So said, I think that the manuscript can be now accepted for publication.

Author Response

Dear Reviewer,

    Thank you for your carefulness and conscientiousness. Your suggestions are really valuable and helpful for revising and improving our paper. According to your suggestions, we have made some revisions and prepared a cover letter to explain the details of the revisions. We look forward to hearing from you at your earliest convenience.

Sincerely yours,

Dingquan Liu  PhD, Prof.
